# Adaptive Anonymity via $b$-Matching

**Krzysztof Choromanski**
Columbia University
kmc2178@columbia.edu

**Tony Jebara**
Columbia University
tj2008@columbia.edu

**Kui Tang**
Columbia University
kt2384@columbia.edu

## Abstract

The adaptive anonymity problem is formalized where each individual shares their data along with an integer value to indicate their personal level of desired privacy. This problem leads to a generalization of $k$-anonymity to the $b$-matching setting. Novel algorithms and theory are provided to implement this type of anonymity. The relaxation achieves better utility, admits theoretical privacy guarantees that are as strong, and, most importantly, accommodates a variable level of anonymity for each individual. Empirical results confirm improved utility on benchmark and social data-sets.

## 1   Introduction

In many situations, individuals wish to share their personal data for machine learning applications and other exploration purposes. If the data contains sensitive information, it is necessary to protect it with privacy guarantees while maintaining some notion of data utility [18, 2, 24]. There are various definitions of privacy. These include $k$-anonymity [19], $l$-diversity [16], $t$-closeness [14] and differential[1] privacy [3, 22]. All these privacy guarantees fundamentally treat each contributed datum about an individual equally. However, the acceptable anonymity and comfort-level of each individual in a population can vary widely. This article explores the adaptive anonymity setting and shows how to generalize the $k$-anonymity framework to handle it. Other related approaches have been previously explored [20, 21, 15, 5, 6, 23] yet herein we contribute novel efficient algorithms and formalize precise privacy guarantees. Note also that there are various definitions of utility. This article focuses on the use of *suppression* since it is well-formalized. Therein, we hide certain values in the data-set by replacing them with a $*$ symbol (fewer $*$ symbols indicate higher utility). The overall goal is to maximize utility while preserving each individual's level of desired privacy.

This article is organized as follows. § 2 formalizes the adaptive anonymity problem and shows how $k$-anonymity does not handle it. This leads to a relaxation of $k$-anonymity into symmetric and asymmetric bipartite regular compatibility graphs. § 3 provides algorithms for maximizing utility under these relaxed privacy criteria. § 4 provides theorems to ensure the privacy of these relaxed criteria for uniform anonymity as well as for adaptive anonymity. § 5 shows experiments on benchmark and social data-sets. Detailed proofs are provided in the Supplement.

## 2   Adaptive anonymity and necessary relaxations to k-anonymity

The adaptive anonymity problem considers a data-set $\mathbf{X} \in \mathbb{Z}^{n \times d}$ consisting of $n \in \mathbb{N}$ observations $\{\mathbf{x}_1, \ldots, \mathbf{x}_n\}$ each of which is a $d$-dimensional discrete vector, in other words, $\mathbf{x}_i \in \mathbb{Z}^d$. Each user $i$ contributes an observation vector $\mathbf{x}_i$ which contains discrete attributes pertaining to that user[2]. Furthermore, each user $i$ provides an adaptive anonymity parameter $\delta_i \in \mathbb{N}$ they desire to keep when the database is released. Given such a data-set and anonymity parameters, we wish to output an obfuscated data-set denoted by $\mathbf{Y} \in \{\mathbb{Z} \cup *\}^{n \times d}$ which consists of vectors $\{\mathbf{y}_1, \ldots, \mathbf{y}_n\}$ where

$\mathbf{y}_i(k) \in \{\mathbf{x}_i(k), *\}$. The star symbol $*$ indicates that the $k$'th attribute has been masked in the $i$'th user-record. We say that vector $\mathbf{x}_i$ is *compatible* with vector $\mathbf{y}_j$ if $\mathbf{x}_i(k) = \mathbf{y}_j(k)$ for all elements of $\mathbf{y}_j(k) \neq *$. The goal of this article is to create a $\mathbf{Y}$ which contains a minimal number of $*$ symbols such that each entry $\mathbf{y}_i$ of $\mathbf{Y}$ is *compatible* with at least $\delta_i$ entries of $\mathbf{X}$ and vice-versa.

The most pervasive method for anonymity in the released data is the $k$-anonymity method [19, 1]. However, it is actually more constraining than the above desiderata. If all users have the same value $\delta_i = k$, then $k$-anonymity suppresses data in the database such that, for each user's data vector in the released (or anonymized) database, there are at least $k - 1$ identical copies in the released database. The existence of copies is used by $k$-anonymity to justify some protection to attack.

We will show that the idea of $k - 1$ copies can be understood as forming a compatibility graph between the original database and the released database which is composed of several fully-connected $k$-cliques. However, rather than guaranteeing copies or cliques, the anonymity problem can be relaxed into a $k$-regular compatibility to achieve nearly identical resilience to attack. More interestingly, this relaxation will naturally allow users to select different $\delta_i$ anonymity values or degrees in the compatibility graph and allow them to achieve their desired personal protection level.

Why can't $k$-anonymity handle heterogeneous anonymity levels $\delta_i$? Consider the case where the population contains many liberal users with very low anonymity levels yet one single paranoid user (user $i$) wants to have a maximal anonymity with $\delta_i = n$. In the $k$-anonymity framework, that user will require $n - 1$ identical copies of his data in the released database. Thus, a single paranoid user will destroy all the information of the database which will merely contain completely redundant vectors. We will propose a $b$-matching relaxation to $k$-anonymity which prevents this degeneracy since it does not merely handle compatibility queries by creating copies in the released data.

While $k$-anonymity is not the only criterion for privacy, there are situations in which it is sufficient as illustrated by the following scenario. First assume the data-set $\mathbf{X}$ is associated with a set of identities (or usernames) and $\mathbf{Y}$ is associated with a set of keys. A key may be the user's password or some secret information (such as their DNA sequence). Represent the usernames and keys using integers $x_1, \ldots, x_n$ and $y_1, \ldots, y_n$, respectively. Username $x_i \in \mathbb{Z}$ is associated with entry $\mathbf{x}_i$ and key $y_j \in \mathbb{Z}$ is associated with entry $\mathbf{y}_j$. Furthermore, assume that these usernames and keys are diverse, unique and independent of their corresponding attributes. These $x$ and $y$ values are known as the *sensitive* attributes and the entries of $\mathbf{X}$ and $\mathbf{Y}$ are the *non-sensitive* attributes [16]. We aim to release an obfuscated database $\mathbf{Y}$ and its keys with the possibility that an adversary may have access to all or a subset of $\mathbf{X}$ and the identities.

The goal is to ensure that the success of an attack (using a username-key pair) is low. In other words, the attack succeeds with probability no larger than $1/\delta_i$ for a user which specified $\delta_i \in \mathbb{N}$. Thus, the attack we seek to protect against is the use of the data to match usernames to keys (rather than attacks in which additional non-sensitive attributes about a user are discovered). In the uniform $\delta_i$ setting, $k$-anonymity guarantees that a single one-time attack using a single username-key pair succeeds with probability at most $1/k$. In the extreme case, it is easy to see that replacing all of $\mathbf{Y}$ with $*$ symbols will result in an attack success probability of $1/n$ if the adversary attempts a single random attack-pair (username and key). Meanwhile, releasing a database $\mathbf{Y} = \mathbf{X}$ with keys could allow the adversary to succeed with an initial attack with probability 1.

We first assume that all degrees $\delta_i$ are constant and set to $\delta$ and discuss how the proposed $b$-matching privacy output subtly differs from standard $k$-anonymity [19]. First, define *quasi-identifiers* as sets of attributes like gender and age that can be linked with external data to uniquely identify an individual in the population. The $k$-anonymity criterion says that a data-set such as $\mathbf{Y}$ is protected against linking attacks that exploit quasi-identifiers if every element is indistinguishable from at least $k - 1$ other elements with respect to *every set* of quasi-identifier attributes. We will instead use a *compatibility graph* $G$ to more precisely characterize how elements are indistinguishable in the data-sets and which entries of $\mathbf{Y}$ are *compatible* with entries in the original data-set $\mathbf{X}$. The graph places edges between entries of $\mathbf{X}$ which are compatible with entries of $\mathbf{Y}$. Clearly, $G$ is an undirected bipartite graph containing two equal-sized partitions (or color-classes) of nodes $A$ and $B$ each of cardinality $n$ where $A = \{a_1, \ldots, a_n\}$ and $B = \{b_1, \ldots, b_n\}$. Each element of $A$ is associated with an entry of $\mathbf{X}$ and each element of $B$ is associated with an entry of $\mathbf{Y}$. An edge $e = (i, j) \in G$ that is adjacent to a node in $A$ and a node in $B$ indicates that the entries $\mathbf{x}_i$ and $\mathbf{y}_j$ are compatible. The absence of an edge means nothing: entries are either *compatible* or *not compatible*.

For $\delta_i = \delta$, $b$-matching produces $\delta$-regular bipartite graphs $G$ while $k$-anonymity produces $\delta$-regular *clique*-bipartite graphs[3] defined as follows.

**Definition 2.1** *Let $G(A, B)$ be a bipartite graph with color classes: $A, B$ where $A = \{a_1, ..., a_n\}, B = \{b_1, ..., b_n\}$. We call a $k$-regular bipartite graph $G(A, B)$ a clique-bipartite graph if it is a union of pairwise disjoint and nonadjacent complete $k$-regular bipartite graphs.*

Denote by $\mathcal{G}_b^{n,\delta}$ the family of $\delta$-regular bipartite graphs with $n$ nodes. Similarly, denote by $\mathcal{G}_k^{n,\delta}$ the family of $\delta$-regular graphs clique-bipartite graphs. We will also denote by $\mathcal{G}_s^{n,\delta}$ the family of symmetric $b$-regular graphs using the following definition of symmetry.

**Definition 2.2** *Let $G(A, B)$ be a bipartite graph with color classes: $A, B$ where $A = \{a_1, ..., a_n\}, B = \{b_1, ..., b_n\}$. We say that $G(A, B)$ is symmetric if the existence of an edge $(a_i, b_j)$ in $G(A, B)$ implies the existence of an edge $(a_j, b_i)$, where $1 \leq i, j \leq n$.*

For values of $n$ that are not trivially small, it is easy to see that the graph families satisfy $\mathcal{G}_k^{n,\delta} \subseteq \mathcal{G}_s^{n,\delta} \subseteq \mathcal{G}_b^{n,\delta}$. This holds since symmetric $\delta$-regular graphs are $\delta$-regular with the additional symmetry constraint. Clique-bipartite graphs are $\delta$-regular graphs constrained to be clique-bipartite and the latter property automatically yields symmetry.

This article introduces graph families $\mathcal{G}_b^{n,\delta}$ and $\mathcal{G}_s^{n,\delta}$ to enforce privacy since these are relaxations of the family $\mathcal{G}_k^{n,b}$ as previously explored in $k$-anonymity research. These relaxations will achieve better utility in the released database. Furthermore, they will allow us to permit adaptive anonymity levels across the users in the database. We will drop the superscripts $n$ and $\delta$ whenever the meaning is clear from the context. Additional properties of these graph families will be formalized in § 4 but we first informally illustrate how they are useful in achieving data privacy.

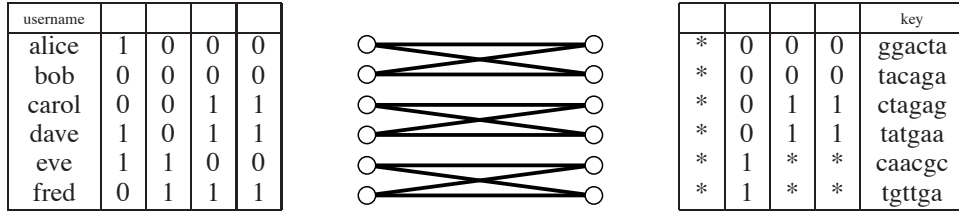

Figure 1: Traditional $k$-anonymity (in $\mathcal{G}_k$) for $n = 6$, $d = 4$, $\delta = 2$ achieves $\#(*) = 10$. Left to right: usernames with data $(x, \mathbf{X})$, compatibility graph $(G)$ and anonymized data with keys $(\mathbf{Y}, y)$.

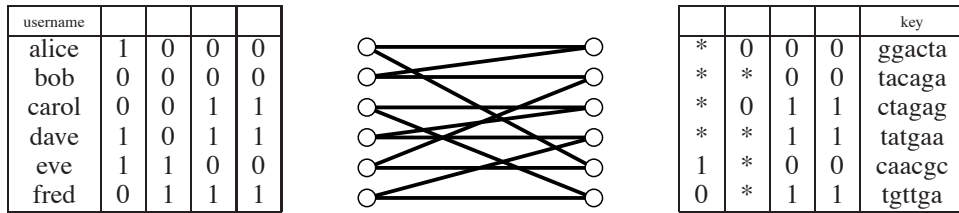

Figure 2: The $b$-matching anonymity (in $\mathcal{G}_b$) for $n = 6$, $d = 4$, $\delta = 2$ achieves $\#(*) = 8$. Left to right: usernames with data $(x, \mathbf{X})$, compatibility graph $(G)$ and anonymized data with keys $(\mathbf{Y}, y)$.

In figure 1, we see an example of $k$-anonymity with a graph from $\mathcal{G}_k$. Here each entry of the anonymized data-set $\mathbf{Y}$ appears $k = 2$ times (or $\delta = 2$). The compatibility graph shows 3 fully connected cliques since each of the $k$ copies in $\mathbf{Y}$ has identical entries. By brute force exploration

we find that the minimum number of stars to achieve this type of anonymity is $\#(*) = 10$. Moreover, since this problem is NP-hard [17], efficient algorithms rarely achieve this best-possible utility (minimal number of stars).

Next, consider figure 2 where we have achieved superior utility by only introducing $\#(*) = 8$ stars to form $\mathbf{Y}$. The compatibility graph is at least $\delta = 2$-regular. It was possible to find a smaller number of stars since $\delta$-regular bipartite graphs are a relaxation of $k$-clique graphs as shown in figure 1. Another possibility (not shown in the figures) is a symmetric version of figure 2 where nodes on the left hand side and nodes on the right hand side have a symmetric connectivity. Such an intermediate solution (since $\mathcal{G}_k \subset \mathcal{G}_s \subset \mathcal{G}_b$) should potentially achieve $\#(*)$ between 8 and 10.

It is easy to see why all graphs have to have a minimum degree of $\delta$ at least (i.e. must contain a $\delta$-regular graph). If one of the nodes has a degree of 1, then the adversary will know the key (or the username) for that node with certainty. If each node has degree $\delta$ or larger, then the adversary will have probability at most $1/\delta$ of choosing the correct key (or username) for any random victim.

We next describe algorithms which accept $\mathbf{X}$ and integers $\delta_1, \ldots, \delta_n$ and output $\mathbf{Y}$ such that each entry $i$ in $\mathbf{Y}$ is *compatible* with at least $\delta_i$ entries in $\mathbf{X}$ and vice-versa. These algorithms operate by finding a graph in $\mathcal{G}_b$ or $\mathcal{G}_s$ and achieve similar protection as $k$-anonymity (which finds a graph in the most restrictive family $\mathcal{G}_k$ and therefore requires more stars). We provide a theoretical analysis of the topology of $G$ in these two new families to show resilience to single and sustained attacks from an all-powerful adversary.

## 3    Approximation algorithms

While the $k$-anonymity suppression problem is known to be NP-hard, a polynomial time method with an approximation guarantee is the *forest* algorithm [1] which has an approximation ratio of $3k - 3$. In practice, though, the forest algorithm is slow and achieves poor utility compared to clustering methods [10]. We provide an algorithm for the $b$-matching anonymity problem with approximation ratio of $\delta$ and runtime of $\mathrm{O}(\delta m \sqrt{n})$ where $n$ is the number of users in the data, $\delta$ is the largest anonymity level in $\{\delta_1, \ldots, \delta_n\}$ and $m$ is the number of edges to explore (in the worst case with no prior knowledge, we have $m = \mathrm{O}(n^2)$ edges between all possible users). One algorithm solves for minimum weight bipartite $b$-matchings which is easy to implement using linear programming, max-flow methods or belief propagation in the bipartite case [9, 11]. The other algorithm uses a general non-bipartite solver which involves Blossom structures and requires $\mathrm{O}(\delta mn \log(n))$ time[8, 9, 13]. Fortunately, minimum weight general matching has recently been shown to require only $\mathrm{O}(m\epsilon^{-1} \log \epsilon^{-1})$ time to achieve a $(1 - \epsilon)$ approximation [7].

First, we define two quantities of interest. Given a graph $G$ with adjacency matrix $\mathbf{G} \in \mathbb{B}^{n \times n}$ and a data-set $\mathbf{X}$, the Hamming error is defined as $h(G) = \sum_i \sum_j \mathbf{G}_{ij} \sum_k (\mathbf{X}_{ik} \neq \mathbf{X}_{jk})$. The number of stars to achieve $G$ is $s(G) = nd - \sum_i \sum_k \prod_j (1 - \mathbf{G}_{ij}(\mathbf{X}_{ik} \neq \mathbf{X}_{jk}))$.

Recall $\mathcal{G}_b$ is the family of regular bipartite graphs. Let $\min_{G \in \mathcal{G}_b} s(G)$ be the minimum number of stars (or suppressions) that one can place in $\mathbf{Y}$ while keeping the entries in $\mathbf{Y}$ compatible with at least $\delta$ entries in $\mathbf{X}$ and vice-versa. We propose the following polynomial time algorithm which, in its first iteration, minimizes $h(G)$ over the family $\mathcal{G}_b$ and then iteratively minimizes a variational upper bound [12] on $s(G)$ using a weighted version of the Hamming distance.

---

**Algorithm 1** variational bipartite $b$-matching

| |
|---|
| Input $\mathbf{X} \in \mathbb{Z}^{n \times d}$, $\delta_i \in \mathbb{N}$ for $i \in \{1, \ldots, n\}$, $\varepsilon > 0$ and initialize $\mathbf{W} \in \mathbb{R}^{n \times d}$ to the all 1s matrix |
| While not converged { |
|     Set $\hat{\mathbf{G}} = \arg\min_{\mathbf{G} \in \mathbb{B}^{n \times n}} \sum_{ij} \mathbf{G}_{ij} \sum_k \mathbf{W}_{ik}(\mathbf{X}_{ik} \neq \mathbf{X}_{jk})$   s.t. $\sum_j \mathbf{G}_{ij} = \sum_j \mathbf{G}_{ji} \geq \delta_i$ |
|     For all $i$ and $k$ set $\mathbf{W}_{ik} = \exp\left(\sum_j \hat{\mathbf{G}}_{ij}(\mathbf{X}_{ik} \neq \mathbf{X}_{jk}) \ln \frac{\varepsilon}{1+\varepsilon}\right)$      } |
| For all $i$ and $k$ set $\mathbf{Y}_{ik} = *$ if $\hat{\mathbf{G}}_{ij} = 1$ and $\mathbf{X}_{jk} \neq \mathbf{X}_{ik}$ for any $j$ |
| Choose random permutation $M$ as matrix $\mathbf{M} \in \mathbb{B}^{n \times n}$ and output $\mathbf{Y}_{public} = \mathbf{MY}$ |

---

We can further restrict the $b$-matching solver such that the graph $G$ is symmetric with respect to both the original data $\mathbf{X}$ and the obfuscated data $\mathbf{Y}$. To do so, we require that $\mathbf{G}$ is a symmetric matrix. This will produce a graph $G \in \mathcal{G}_s$. In such a situation, the value of $\hat{\mathbf{G}}$ is recovered by a general

unipartite $b$-matching algorithm rather than a bipartite $b$-matching program. Thus, the set of possible output solutions is strictly smaller (the bipartite formulation relaxes the symmetric one).

---

**Algorithm 2** variational symmetric $b$-matching

| |
|---|
| Input $\mathbf{X} \in \mathbb{Z}^{n \times d}, \delta_i \in \mathbb{N}$ for $i \in \{1, \dots, n\}, \varepsilon > 0$ and initialize $\mathbf{W} \in \mathbb{R}^{n \times d}$ to the all 1s matrix |
| While not converged { |
| $\quad$ Set $\hat{\mathbf{G}} = \arg\min_{\mathbf{G} \in \mathbb{B}^{n \times n}} \sum_{ij} \mathbf{G}_{ij} \sum_k \mathbf{W}_{ik}(\mathbf{X}_{ik} \neq \mathbf{X}_{jk})$ s.t. $\sum_j \mathbf{G}_{ij} \geq \delta_i$, $\mathbf{G}_{ij} = \mathbf{G}_{ji}$ |
| $\quad$ For all $i$ and $k$ set $\mathbf{W}_{ik} = \exp\left(\sum_j \hat{\mathbf{G}}_{ij}(\mathbf{X}_{ik} \neq \mathbf{X}_{jk}) \ln \frac{\varepsilon}{1+\varepsilon}\right)$ } |
| For all $i$ and $k$ set $\mathbf{Y}_{ik} = *$ if $\hat{\mathbf{G}}_{ij} = 1$ and $\mathbf{X}_{jk} \neq \mathbf{X}_{ik}$ for any $j$ |
| Choose random permutation $M$ as matrix $\mathbf{M} \in \mathbb{B}^{n \times n}$ and output $\mathbf{Y}_{public} = \mathbf{MY}$ |

---

**Theorem 1** *For $\delta_i \leq \delta$, iteration #1 of algorithm 1 finds $\hat{G}$ such that $s(\hat{G}) \leq \delta \min_{G \in \mathcal{G}_b} s(G)$.*

**Theorem 2** *Each iteration of algorithm 1 monotonically decreases $s(\hat{G})$.*

Theorem 1 and 2 apply to both algorithms. Both algorithms[4] manipulate a bipartite regular graph $G(A, B)$ containing the true matching $\{(a_1, b_1), \dots, (a_n, b_n)\}$. However, they ultimately release the data-set $\mathbf{Y}_{public}$ after randomly shuffling $\mathbf{Y}$ according to some matching or permutation $M$ which hides the true matching. The random permutation or matching $M$ can be represented as a matrix $\mathbf{M} \in \mathbb{B}^{n \times n}$ or as a function $\sigma : \{1, \dots, n\} \to \{1, \dots, n\}$. We now discuss how an adversary can attack privacy by recovering this matching or parts of it.

## 4 Privacy guarantees

We now characterize the anonymity provided by a compatibility graph $G \in \mathcal{G}_b$ (or $G \in \mathcal{G}_s$) under several attack models. The goal of the adversary is to correctly match people to as many records as possible. In other words, the adversary wishes to find the random matching $M$ used in the algorithms (or parts of $M$) to connect the entries of $\mathbf{X}$ to the entries of $\mathbf{Y}_{public}$ (assuming the adversary has stolen $\mathbf{X}$ and $\mathbf{Y}_{public}$ or portions of them). More precisely, we have a bipartite graph $G(A, B)$ with color classes $A, B$, each of size $n$. Class $A$ corresponds to $n$ usernames and class $B$ to $n$ keys. Each username in $A$ is matched to its key in $B$ through some unknown matching $M$.

We consider the model where the graph $G(A, B)$ is $\delta$-regular, where $\delta \in \mathbb{N}$ is a parameter chosen by the publisher. The latter is especially important if we are interested in guaranteeing different levels of privacy for different users and allowing $\delta$ to vary with the user's index $i$.

Sometimes it is the case that the adversary has some additional information and at the very beginning knows some complete records that belong to some people. In graph-theoretic terms, the adversary thus knows parts of the hidden matching $M$ in advance. Alternatively, the adversary may have come across such additional information through *sustained attack* where previous attempts revealed the presence or absence of an edge. We are interested in analyzing how this extra knowledge can help him further reveal other edges of the matching. We aim to show that, for some range of the parameters of the bipartite graphs, this additional knowledge does not help him much. We will compare the resilience to attack relative to the resilience of $k$-anonymity. We say that a person $v$ is $k$-anonymous if his or her real data record can be confused with at least $k - 1$ records from different people. We first discuss the case of single attacks and then discuss sustained attacks.

### 4.1 One-Time Attack Guarantees

Assume first that the adversary has no extra information about the matching and performs a one-time attack. Then, lemma 4.1 holds which is a direct implication of lemma 4.2.

**Lemma 4.1** *If $G(A, B)$ is an arbitrary $\delta$-regular graph and the adversary does not know any edges of the matching he is looking for then every person is $\delta$-anonymous.*

**Lemma 4.2** *Let $G(A,B)$ be a $\delta$-regular bipartite graph. Then for every edge $e$ of $G(A,B)$ there exists a perfect matching in $G(A,B)$ that uses $e$.*

The result does not assume any structure in the graph beyond its $\delta$-regularity. Thus, for a single attack, $b$-matching anonymity (symmetric or asymmetric) is equivalent to $k$-anonymity when $b = k$.

**Corollary 4.1** *Assume the bipartite graph $G(A,B)$ is either $\delta$-regular, symmetric $\delta$-regular or clique-bipartite and $\delta$-regular. An adversary attacking $G$ once succeeds with probability $\leq 1/\delta$.*

## 4.2 Sustained Attack on $k$-Cliques

Now consider the situation of sustained attacks or attacks with prior information. Here, the adversary may know $c \in \mathbb{N}$ edges in $M$ a priori by whatever means (previous attacks or through side information). We begin by analyzing the resilience of $k$-anonymity where $G$ is a cliques-structured graph. In the clique-bipartite graph, even if the adversary knows some edges of the matching (but not too many) then there still is hope of good anonymity for all people. The anonymity of every person decreases from $\delta$ to at least $(\delta - c)$. So, for example, if the adversary knows in advance $\frac{\delta}{2}$ edges of the matching then we get the same type of anonymity for every person as for the model with two times smaller degree in which the adversary has no extra knowledge. So we will be able to show the following:

**Lemma 4.3** *If $G(A,B)$ is clique-bipartite $\delta$-regular graph and the adversary knows in advance $c$ edges of the matching then every person is $(\delta - c)$-anonymous.*

The above is simply a consequence of the following lemma.

**Lemma 4.4** *Assume that $G(A,B)$ is clique-bipartite $\delta$-regular graph. Denote by $M$ some perfect matching in $G(A,B)$. Let $C$ be some subset of the edges of $M$ and let $c = |C|$. Fix some vertex $v \in A$ not matched in $C$. Then there are at least $(\delta - c)$ edges adjacent to $v$ such that, for each of these edges $e$, there exists some perfect matching $M^e$ in $G(A,B)$ that uses both $e$ and $C$.*

**Corollary 4.2** *Assume graph $G(A,B)$ is a clique-bipartite and $\delta$-regular. Assume that the adversary knows in advance $c$ edges of the matching. The adversary selects uniformly at random a vertex the privacy of which he wants to break from the set of vertices he does not know in advance. Then he succeeds with probability at most $\frac{1}{\delta - c}$.*

We next show that $b$-matchings achieve comparable resilience under sustained attack.

## 4.3 Sustained attack on asymmetric bipartite $b$-matching

We now consider the case where we do not have a graph $G(A,B)$ which is clique-bipartite but rather is only $\delta$-regular and potentially asymmetric (as returned by algorithm 1).

**Theorem 4.1** *Let $G(A,B)$ be a $\delta$-regular bipartite graph with color classes: $A$ and $B$. Assume that $|A| = |B| = n$. Denote by $M$ some perfect matching $M$ in $G(A,B)$. Let $C$ be some subset of the edges of $M$ and let $c = |C|$. Take some $\xi \geq c$. Denote $n' = n - c$. Fix any function $\phi : N \to R$ satisfying $\forall_\delta (\xi \sqrt{2\delta + \frac{1}{4}} < \phi(\delta) < \delta)$. Then for all but at most*

$$\eta = \frac{2c\delta^2 n' \xi (1 + \frac{\phi(\delta) + \sqrt{\phi^2(\delta) - 2\xi^2\delta}}{2\xi\delta})}{\phi^3(\delta)(1 + \sqrt{1 - \frac{2\xi^2\delta}{\phi^2(\delta)}})(\frac{1}{\xi} - \frac{c}{\phi(\delta)} + \frac{\delta(1 - \frac{c}{\xi})}{\phi(\delta)})} + \frac{c\delta}{\phi(\delta)} \text{ vertices } v \in A \text{ not matched in } C \text{ the following}$$

*holds:* The size of the set of edges $e$ adjacent to $v$ and having the additional property that there exists some perfect matching $M^v$ in $G(A,B)$ that uses $e$ and edges from $C$ is: at least $(\delta - c - \phi(\delta))$.

Essentially, theorem 4.1 says that all but at most a small number $\eta$ of people are $(\delta - c - \phi(\delta))$-anonymous for every $\phi$ satisfying: $c\sqrt{2\delta + \frac{1}{4}} < \phi(\delta) < \delta$ if the adversary knows in advance $c$ edges of the matching. For example, take $\phi(\delta) := \theta\delta$ for $\theta \in (0,1)$. Fix $\xi = c$ and assume that the adversary knows in advance at most $\delta^{\frac{1}{4}}$ edges of the matching. Then, using the formula from

theorem 4.1, we obtain that (for $n$ large enough) all but at most $\frac{4n'}{\delta^{\frac{1}{4}}\theta^3} + \frac{\delta^{\frac{1}{4}}}{\theta}$ people from those that the adversary does not know in advance are $((1-\theta)\delta - \delta^{\frac{1}{4}})$-anonymous. So if $\delta$ is large enough then all but approximately a small fraction $\frac{4}{\delta^{\frac{1}{4}}\theta^3}$ of all people not known in advance are almost $(1-\theta)\delta$-anonymous.

Again take $\phi(\delta) := \theta\delta$ where $\theta \in (0,1)$. Take $\xi = 2c$. Next assume that $1 \le c \le \min(\frac{\delta}{4}, \delta(1-\theta-\theta^2))$. Assume that the adversary selects uniformly at random a person to attack. Our goal is to find an upper bound on the probability he succeeds. Then, using theorem 4.1, we can conclude that all but at most $Fn'$ people whose records are not known in advance are $((1-\theta)\delta - c)$-anonymous for $F = \frac{33c^2}{\theta^2\delta}$. The probability of success is at most: $F + (1-F)\frac{1}{(1-\theta)\delta-c}$. Using the expression on $F$ that we have and our assumptions, we can conclude that the probability we are looking for is at most $\frac{34c^2}{\theta^2\delta}$. Therefore we have:

**Theorem 4.2** *Assume graph $G(A,B)$ is $\delta$-regular and the adversary knows in advance $c$ edges of the matching, where $c$ satisfies: $1 \le c \le \min(\frac{\delta}{4}, \delta(1-\theta-\theta^2))$. The adversary selects uniformly at random a vertex the privacy of which he wants to break from those that he does not know in advance. Then he succeeds with probability at most $\frac{34c^2}{\theta^2\delta}$.*

### 4.4 Sustained attack on symmetric $b$-matching with adaptive anonymity

We now consider the case where the graph is not only $\delta$-regular but also symmetric as defined in definition 2.2 and as recovered by algorithm 2. Furthermore, we consider the case where we have varying values of $\delta_i$ for each node since some users want higher privacy than others. It turns out that if the corresponding bipartite graph is symmetric (we define this term below) we can conclude that each user is $(\delta_i - c)$-anonymous, where $\delta_i$ is the degree of a vertex associated with the user of the bipartite matching graph. So we get results completely analogous to those for the much simpler models described before. We will use a slightly more elaborate definition of symmetric[5], however, since this graph has one if its partitions permuted by a random matching (the last step in both algorithms before releasing the data).

**Definition 4.1** *Let $G(A,B)$ be a bipartite graph with color classes: $A, B$ and matching $M = \{(a_1, b_1), ...(a_n, b_n)\}$, where $A = \{a_1, ..., a_n\}, B = \{b_1, ..., b_n\}$. We say that $G(A,B)$ is symmetric with respect to $M$ if the existence of an edge $(a_i, b_j)$ in $G(A,B)$ implies the existence of an edge $(a_j, b_i)$, where $1 \le i, j \le n$.*

From now on, the matching $M$ with respect to which $G(A,B)$ is symmetric is a canonical matching of $G(A,B)$. Assume that $G(A,B)$ is symmetric with respect to its canonical matching $M$ (it does not need to be a clique-bipartite graph). In such a case, we will prove that, if the adversary knows in advance $c$ edges of the matching, then every person from the class $A$ of degree $\delta_i$ is $(\delta_i - c)$-anonymous. So we obtain the *same type of anonymity* as in a clique-bipartite graph (see: lemma 4.3).

**Lemma 4.5** *Assume that $G(A,B)$ is a bipartite graph, symmetric with respect to its canonical matching $M$. Assume furthermore that the adversary knows in advance $c$ edges of the matching. Then every person that he does not know in advance is $(\delta_i - c)$-anonymous, where $\delta_i$ is a degree of the related vertex of the bipartite graph.*

As a corollary, we obtain *the same* privacy guarantees in the symmetric case as the $k$-cliques case.

**Corollary 4.3** *Assume bipartite graph $G(A,B)$ is symmetric with respect to its canonical matchings $M$. Assume that the adversary knows in advance $c$ edges of the matching. The adversary selects uniformly at random a vertex the privacy of which he wants to break from the set of vertices he does not know in advance. Then he succeeds with probability at most $\frac{1}{\delta_i - c}$, where $\delta_i$ is a degree of a vertex of the matching graph associated with the user.*

In summary, the symmetric case is *as resilient* to sustained attack as the cliques-bipartite case, the usual one underlying $k$-anonymity if we set $\delta_i = \delta = k$ everywhere. The adversary succeeds with probability at most $1/(\delta_i - c)$. However, the asymmetric case is potentially weaker and the adversary can succeed with probability at most $\frac{34c^2}{\theta^2 \delta}$. Interestingly, in the symmetric case with variable $\delta_i$ degrees, however, we can provide guarantees that are just as good without forcing all individuals to agree on a common level of anonymity.

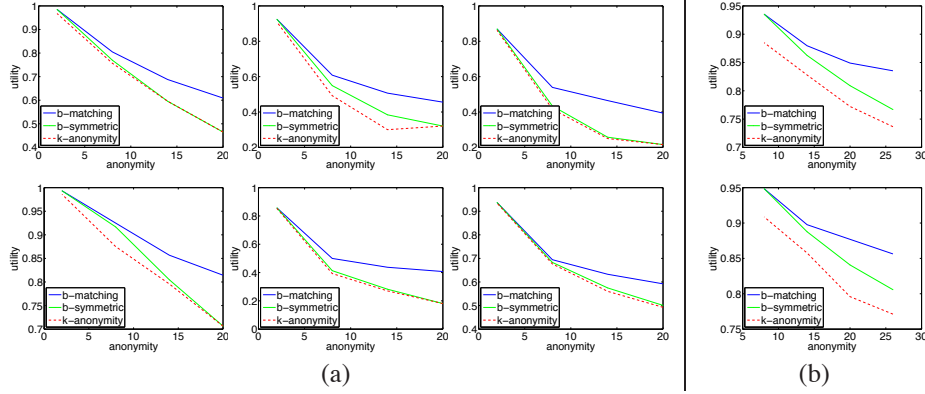

(a)                                                                                      (b)

Figure 3: Utility $(1 - \frac{\#(*)}{nd})$ versus anonymity on (a) Bupa $(n = 344, d = 7)$, Wine $(n = 178, d = 14)$, Heart $(n = 186, d = 23)$, Ecoli $(n = 336, d = 8)$, and Hepatitis $(n = 154, d = 20)$ and Forest Fires $(n = 517, d = 44)$ data-sets and (b) CalTech University Facebook $(n = 768, d = 101)$ and Reed University Facebook $(n = 962, d = 101)$ data-sets.

# 5   Experiments

We compared algorithms 1 and 2 against an agglomerative clustering competitor (optimized to minimize stars) which is known to outperform the *forest* method [10]. Agglomerative clustering starts with singleton clusters and keeps unifying the two closest clusters with smallest increase in stars until clusters grow to a size at least $k$. Both algorithms release data with suppressions to achieve a desired constant anonymity level $\delta$. For our algorithms, we swept values of $\varepsilon$ in $\{2^{-1}, 2^{-2}, \ldots, 2^{-10}\}$ from largest to smallest and chose the solution that produced the least number of stars. Furthermore, we warm-started the symmetric algorithm with the star-pattern solution of the asymmetric algorithm to make it converge more quickly. We first explored six standard data-sets from UCI http://archive.ics.uci.edu/ml/ in the uniform anonymity setting. Figure 3(a) summarizes the results where utility is plotted against $\delta$. Fewer stars imply greater utility and larger $\delta$ implies higher anonymity. We discretized each numerical dimension in a data-set into a binary attribute by finding all elements above and below the median and mapped categorical values in the data-sets into a binary code (potentially increasing the dimensionality). Algorithms 1 achieved significantly better utility for any given fixed constant anonymity level $\delta$ while algorithm 2 achieved a slight improvement. We next explored Facebook social data experiments where each user has a different level of desired anonymity and has 7 discrete profile attributes which were binarized into $d = 101$ dimensions. We used the number of friends $f_i$ a user has to compute their desired anonymity level (which decreases as the number of friends increases). We set $F = \max_{i=1,\ldots n} \lfloor \log f_i \rfloor$ and, for each value of $\delta$ in the plot, we set user $i$'s privacy level to $\delta_i = \delta - (F - \lfloor \log f_i \rfloor)$. Figure 3(b) summarizes the results where utility is plotted against $\delta$. Since the $k$-anonymity agglomerative clustering method requires a constant $\delta$ for all users, we set $k = \max_i \delta_i$ in order to have a privacy guarantee. Algorithms 1 and 2 consistently achieved significantly better utility in the adaptive anonymity setting while also achieving the desired level of privacy protection.

# 6   Discussion

We described the adaptive anonymity problem where data is obfuscated to respect each individual user's privacy settings. We proposed a relaxation of $k$-anonymity which is straightforward to implement algorithmically. It yields similar privacy protection while offering greater utility and the ability to handle heterogeneous anonymity levels for each user.

## Footnotes

[1]Differential privacy often requires specifying the data application (e.g. logistic regression) in advance [4].

[2]For instance, a vector can contain a user's gender, race, height, weight, age, income bracket and so on.

[3] Traditional $k$-anonymity releases an obfuscated database of $n$ rows where there are $k$ copies of each row. So, each copy has the same neighborhood. Similarly, the entries of the original database all have to be connected to the same $k$ copies in the obfuscated database. This induces a so-called bipartite clique-connectivity.

[4]It is straightforward to put a different weight on certain suppressions over others if the utility of the data is not uniform for each entry or bit. This done by using an $n \times d$ weight matrix in the optimization. It is also straightforward to handle missing data by allowing initial stars in $\mathbf{X}$ before anonymizing.

[5]A symmetric graph $G(A,B)$ may not remain symmetric according to definition 2.2 if nodes in $B$ are shuffled by a permutation $M$. However, it will still be *symmetric with respect to $M$* according to definition 4.1.

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
