[Supplementary Material · nips13anonSUPPLEMENT.pdf]

# Adaptive Anonymity via $b$-Matching
# Supplementary Material

## Abstract

This supplement contains all necessary detailed proofs and a worst-case theoretical analysis in support of the main article.

## 7    Proof of theorem 1

**Proof 1** *In the first iteration, the algorithm is clearly solving $\hat{G} = \arg\min_{G \in \mathcal{G}_b} h(G)$. Let $G^* = \arg\min_{G \in \mathcal{G}_b} s(G)$. Clearly, the number of stars is less than the Hamming distance $s(G) \le h(G)$ for any $G$. Since $\hat{G}$ is the minimizer of $h(G)$, we have $h(\hat{G}) \le h(G^*)$. Furthermore, it is easy to show for $\delta$-regular graphs that $h(G) \le \delta s(G)$. Combining yields $s(\hat{G}) \le h(\hat{G}) \le h(G^*) \le \delta s(G^*) = \delta \min_{G \in \mathcal{G}_b} s(G)$.*

## 8    Proof of theorem 2

**Proof 2** *Create an $\varepsilon$-approximation $\tilde{s}(G)$ to $s(G)$ by adding a tiny $\varepsilon > 0$ to each term in the product*

$$\tilde{s}(G) = nd - \exp \ln \sum_{ik} \frac{\mathbf{W}_{ik}}{\mathbf{W}_{ik}} \prod_j \left(1 + \varepsilon - \mathbf{G}_{ij}(\mathbf{X}_{ik} \ne \mathbf{X}_{jk})\right)$$

$$\le nd - e^{\sum_{ijk} \mathbf{W}_{ik}\left(\mathbf{G}_{ij}(\mathbf{X}_{ik} \ne \mathbf{X}_{jk}) \ln \frac{\varepsilon}{1+\varepsilon} + \ln(1+\varepsilon)\right) - \sum_{ik} \mathbf{W}_{ik} \ln \mathbf{W}_{ik}}$$

*where we introduced a variational parameter $\mathbf{W} \in \mathbb{Z}^{n \times d}$ s.t. $\sum_{ik} \mathbf{W}_{ik} = 1$ and applied Jensen's inequality. The first step of the "while" loop minimizes the right hand side over $\mathbf{G}$ while the second minimizes over $\mathbf{W}$ (modulo a harmless scaling). Thus, the algorithm minimizes a variational upper bound on $\tilde{s}(G)$ which cannot increase. Since the parameter $G$ is discrete, $\tilde{s}(G)$ must decrease with every iteration or else the algorithm terminates (converges).*

## 9    Proof of lemma 4.2

**Proof 3** *Take some perfect matching $M_1$ in $G(A, B)$ (it exists because of Hall's theorem). If it uses $e$ then we are done. Assume it does not. Delete all edges from $M_1$ from $G(A, B)$ to obtain a $(\delta - 1)$-bipartite graph. Take one of its perfect matchings, say $M_2$. If it uses $e$ then we are done. Otherwise delete edges from $M_2$ and continue. At some point, some perfect matching will use $e$ because, otherwise, we end up with an empty graph (i.e. without edges).*

## 10    Proof of lemma 4.4

**Proof 4** *Denote by $\breve{G} = G(\hat{A}, \hat{B})$ the graph obtained from $G(A, B)$ by deleting vertices of $M$. Obviously it has a perfect matching, namely: $M - C$. In fact $\breve{G}$ is a union of complete bipartite graphs, pairwise disjoint, each with color classes of size at least $(\delta - c)$. Each perfect matching in $\breve{G}$ is a union of perfect matchings of those complete bipartite graphs. Denote by $\breve{G}_v$ a complete bipartite graph of $\breve{G}$ corresponding to vertex $v$. Then obviously for every edge $e$ in $\breve{G}_v$ there is a perfect matching in $\breve{G}_v$ that uses $e$. In $\breve{G}_v$ we have at least $(\delta - c)$ edges adjacent to $v$ and that completes the proof.*

# 11 Proof of theorem 4.1

**Proof 5** *Take perfect matching $M$ and $C \subseteq M$ from the statement of the theorem. For every vertex $v \in A$, denote by $m(v)$ its neighbor in $M$. Denote: $m(V) = \{m(v) : v \in V\}$. Take bipartite graph $\breve{G} = G(\breve{A}, \breve{B})$ with color classes $\breve{A}, \breve{B}$, obtained from $G(A, B)$ by deleting all vertices of $C$. For a vertex $v \in \breve{A}$ and an edge $e$ adjacent to it in $\breve{G}$ we will say that this edge is bad with respect to $v$ if there is no perfect matching in $G(A, B)$ that uses $e$ and all edges from $C$. We will say that a vertex $v \in \breve{A}$ is bad if there are at least $\phi(\delta)$ edges that are bad with respect to $v$. Denote by $x$ the number of bad vertices and by $X$ the set of all bad vertices. We just need to prove that*

$$x \leq \frac{2c^3\delta^2 n'(1 + \frac{\phi(\delta) + \sqrt{\phi^2(\delta) - 2c^2\delta}}{2c\delta})}{\phi^3(\delta)(1 + \sqrt{1 - \frac{2c^2\delta}{\phi^2(\delta)}})(1 - \frac{c^2}{\phi(\delta)})} + \frac{c\delta}{\phi(\delta)}. \text{ Take some bad vertex } v \text{ and some edge } e \text{ which is bad}$$

*with respect to it. Graph $\breve{G}$ obviously has a perfect matching, namely: $M - C$. However from the definition of $e$, it does not have a perfect matching that uses $e$. So the graph $\breve{G}^e = G(\breve{A}^e, \breve{B}^e)$ obtained from $\breve{G}$ by deleting both endpoints of $e$ does not have a perfect matching. But, according to Hall's theorem, that means that in $\breve{G}^e$ there is a subset $S_v^e \subseteq \breve{A}^e$ such that $|N(S_v^e)| < |S_v^e|$, where $N(T)$ denotes the set of neighbors of the vertices from the set $T$. But in $\breve{G}$ we have: $|N(S_v^e)| \geq |S_v^e|$. In fact we can say more: $m(S_v^e) \subseteq N(S_v^e)$ in $\breve{G}$. Therefore it must be the case that an edge $e$ touches a vertex from $m(S_v^e)$ and furthermore $N(S_v^e) = m(S_v^e)$ in $\breve{G}$. Whenever the set $S \subseteq \breve{A}$ satisfies: $N(S) = m(S)$ in $\breve{G}$ we say that it is* closed. *So for for every edge $e$ bad with respect to a vertex $v$ there exists closed set $S_v^e$. Fix some bad vertex $v$ and some set $E$ of its bad edges with size $\phi(\delta)$. Denote $S_v^E = \bigcup_{e \in E} S_v^e$. $S_v^E$ is closed as a sum of closed sets. We also have: $v \notin S_v^E$. Besides every edge from $E$ touches some vertex from $m(S_v^E)$. We say that the set $S$ is $\phi(\delta)$-bad with respect to a vertex $v \in \breve{A} - S$ if it is closed and there are $\phi(\delta)$ bad edges with respect to $v$ that touch $S$. So we conclude that $S_v^E$ is $\phi(\delta)$-bad with respect to $v$. Let $S_v^m$ be the minimal $\phi(\delta)$-bad set with respect to $v$.*

**Lemma 11.1** *Let $v_1, v_2$ be two bad vertices. If $v_2 \in S_{v_1}^m$ then $S_{v_2}^m \subseteq S_{v_1}^m$.*

**Proof 6** *From the fact that $S_{v_1}^m$ is closed we know $\phi(\delta)$ bad edges adjacent to $v_2$ and touching $m(S_{v_2}^m)$ must also touch $m(S_{v_1}^m)$. So those $\phi(\delta)$ edges also touch $m(S_{v_2}^m \bigcap S_{v_1}^m)$. Clearly the set $T = S_{v_2}^m \bigcap S_{v_1}^m$ is closed as an intersection of two closed sets. So from what we know so far we can conclude that it is $\phi(\delta)$-bad with respect to $v_2$. So from the definition of $S_{v_2}^m$ we can conclude that $T = S_{v_2}^m$, so $S_{v_2}^m \subseteq S_{v_1}^m$.*

**Lemma 11.2** *Denote $P = \{S_v^m : v \in X\}$. It is a poset with the ordering induced by the inclusion relation. Then it does not have anti-chains of size larger than $\frac{c\delta}{\phi(\delta)}$.*

**Proof 7** *Take some anti-chain $A = \{S_{v_1}^m, ..., S_{v_l}^m\}$ in $P$. From lemma 11.1 we know that the set $v_1, ..., v_l$ does not intersect $R = S_{v_1}^m \bigcup S_{v_2}^m ... \bigcup S_{v_l}^m$. But $R$ is closed as a sum of closed sets. Assume by contradiction that $l > \frac{c\delta}{\phi(\delta)}$, i.e. $\phi(\delta)l > c\delta$. Now consider the set $D = m(R)$. We will count the number of edges touching $D$ in $G(A, B)$. On the one hand from the fact that $G(A, B)$ is $\delta$-regular we know that this number is exactly $l\delta$. On the other hand we have at least $\phi(\delta)l$ edges ($\phi(\delta)$ bad edges from every $v_i : i = 1, 2, ..., l$) touching $D$. Besides from the fact that $R$ is closed we know that there are at least $l\delta - c\delta$ edges such that each of them is adjacent to some vertex from $R$ and from $m(R)$ (for every vertex from $R$ we have $\delta$ edges in $G(A, B)$ adjacent to it and all but the edges adjacent to some vertices from $C$ must touch $D$; altogether we have at most $c\delta$ edges such that each of them is adjacent to some vertex from $C$ and some vertex from $R$). So summing all those edges we get more than $\delta l$ edges which is a contradiction.*

**Corollary 11.1** *Using Dillworth's lemma about chains and anti-chains in posets and lemma 11.2, we see that the set $P = \{S_v^m : v \in X\}$ has a chain of length at least $\frac{x\phi(\delta)}{c\delta}$.*

*Now take an arbitrary chain of $P = \{S_v^m : v \in X\}$ of length at least $\frac{x\phi(\delta)}{c\delta}$. Denote $L = \{S_{v_1}^m, ..., S_{v_d}^m\}$, where $S_{v_1}^m \subseteq S_{v_2}^m \subseteq ... \subseteq S_{v_d}^m$. So we have $d \geq \frac{x\phi(\delta)}{c\delta}$. Denote: $X_i = S_{v_{i+1}}^m - S_{v_i}^m$ for*

$i = 1, 2, ..., d-1$. Assume that $|X_i| \geq (\xi+1)$. Then $X_i$ contains at least $\xi$ vertices different than $v_i$. Call this set of vertices $C_i$. At least one vertex from $C_i$ must have at most $(\frac{c\delta-\phi(\delta)}{\xi})$ edges adjacent to it and touching $m(S_{v_i}^m)$. Assume not and count the number of edges of $G(A, B)$ with one endpoint in $m(S_{v_i}^m)$. Then we have more than $\xi\frac{c\delta-\phi(\delta)}{\xi}$ such edges adjacent to vertices from $C_i$. Moreover, there are at least $\phi(\delta)$ bad edges adjacent to vertex $v_i$. Finally we have at least $\delta|S_{v_i}^m| - c\delta$ edges like that adjacent to vertices from $S_{v_i}^m$ (the analysis of this last expression is the same as in lemma 11.2). So altogether we have more than $\delta|S_{v_i}^m|$ which is impossible because $G(A, B)$ is $\delta$-regular. So we can conclude that if $|X_i| \geq (\xi+1)$ then $X_i$ contains vertex $x_i$ with at most $(\frac{c\delta-\phi(\delta)}{\xi})$ edges adjacent to it and touching $m(S_{v_i}^m)$. So there are at least $\delta - \frac{c\delta-\phi(\delta)}{\xi}$ edges adjacent to $x_i$ with second endpoints in $B - m(S_{v_i}^m)$. But we also know that $x_i \in S_{v_{i+1}}^m$ and the set $S_{v_{i+1}}^m$ is closed. So at least $\delta - \frac{c\delta-\phi(\delta)}{\xi} - c$ edges adjacent to $x_i$ have second endpoints in $m(S_{v_{i+1}}^m - S_{v_i}^m) = m(X_i)$. But that means that $|m(X_i)| \geq \delta - \frac{c\delta-\phi(\delta)}{\xi} - c$, so $|X_i| \geq \delta - \frac{c\delta-\phi(\delta)}{\xi} - c$. So we can conclude that if $|X_i| \geq (\xi + 1)$ then $X_i \geq \delta - \frac{c\delta-\phi(\delta)}{\xi} - c$. Let's now analyze how many consecutive sets $X_i$ may satisfy $|X_i| \leq \xi$. Assume that sets $X_{i+1}, ..., X_{i+l}$ all have size at most $\xi$. Consider vertices $v_{i+1}, v_{i+2}, ..., v_{i+j}$ and the sets of bad edges $E(v_{i+j})$ related to $v_{i+j}$ and $S_{v_{i+j}}^m$ for $1 \leq j \leq l$. We have: $|E(v_{i+j})| \geq \phi(\delta)$. We will count the number of edges in $G(A, B)$ that touch $m(S_{v_i}^m)$. All edges from $E(v_{i+1})$ satisfy this condition. At least $E(v_{i+2}) - |X_i|$ from $E(v_{i+2})$ satisfy it, at least $E(v_{i+3}) - |X_i| - |X_{i+1}|$ edges from $E(v_{i+3})$, etc. So we have at least $l\phi(\delta) - \xi - 2\xi - ... - (l-1)\xi$ edges from $\bigcup_{j=1}^{l} E(v_{i+j})$ satisfying this condition. Furthermore we have at least $\delta|S_{v_i}^m| - c\delta$ other edges satisfying it (with one endpoint in $S_{v_i}^m$). Because altogether we can't have more than $\delta|S_{v_i}^m|$ edges satisfying the condition, we must have: $l\phi(\delta) - \xi - 2\xi - ... - (l-1)\xi \leq c\delta$. Note that $\xi \geq c$. So we have in particular $l\phi(\delta) - \frac{\xi l^2}{2} \leq \xi\delta$. Solving this quadratic equation we obtain: $l \leq \frac{\phi(\delta)-\sqrt{\phi^2(\delta)-2\xi^2\delta}}{\xi}$ or $l \geq \frac{\phi(\delta)+\sqrt{\phi^2(\delta)-2\xi^2\delta}}{\xi}$. Now assume that we have more than $\frac{\phi(\delta)-\sqrt{\phi^2(\delta)-2\xi^2\delta}}{\xi}$ consecutive $X_i$ of size at most $\xi$. Then let $r$ be the smallest integer greater than $\frac{\phi(\delta)-\sqrt{\phi^2(\delta)-2\xi^2\delta}}{\xi}$. Take $r$ consecutive sets: $X_{i+1}, ..., X_{i+r}$. Then it's easy to check that the condition $\phi(\delta) > \xi\sqrt{2\delta + \frac{1}{4}}$ implies that we have: $\phi(\delta) - \sqrt{\phi^2(\delta) - 2\xi^2\delta} < \xi r < \phi(\delta) + \sqrt{\phi^2(\delta) - 2\xi^2\delta}$. But this is a contradiction according to what we have said so far. Therefore we must have: $l \leq l_0$, where $l_0 = \frac{\phi(\delta)-\sqrt{\phi^2(\delta)-2\xi^2\delta}}{\xi}$. But that means that in the set: $\{X_1, X_2, ..., X_{d-1}\}$ we have at least $\frac{d-1}{l_0+1}$ sets of size at least $\delta - \frac{c\delta-\phi(\delta)}{\xi} - c$. Sets $X_i$ are pairwise disjoint and are taken from the set of size $n' = n - c$. Therefore we have: $\frac{d-1}{l_0+1}(\delta - \frac{c\delta-\phi(\delta)}{\xi} - c) \leq n'$. So we have: $d \leq \frac{n'(l_0+1)}{\delta - \frac{c\delta-\phi(\delta)}{\xi} - c} + 1$. But then using the inequality $d \geq \frac{x\phi(\delta)}{c\delta}$ and substituting in the expression for $l_0$ we complete the proof of the theorem.

## 12   Proof of lemma 4.5

**Proof 8** *Take canonical matching* $M = \{(a_1, b_1), ..., (a_n, b_n)\}$ *of* $G(A, B)$. *Without loss of generality assume that the adversary knows the edges:* $\{(a_1, b_1), ..., (a_c, b_c)\}$. *Write* $C = \{(a_1, b_1), ..., (a_c, b_c)\}$ *and* $m(a_i) = b_i$, *for* $i = 1, 2, ..., n$. *Denote the degree of a vertex* $a_i$ *in* $G(A, B)$ *as* $\delta_i$ *for* $i = 1, 2, ..., n$. *Note that from our assumption about the bipartite graph we know that the degree of a vertex* $b_i$ *is also* $\delta_i$ *for* $i = 1, 2, ..., n$. *For a subset* $S \subseteq A$ *denote* $m(S) = \{m(v) : v \in S\}$. *Take vertex* $v = a_i$ *for* $i > c$. *An edge* $e$ *non-incident with edges from* $C$, *but incident with* $v$ *is a good edge if there exists a perfect matching in* $G(A, B)$ *that uses* $e$ *such that* $C$ *is its sub-matching. It suffices to prove that for any fixed* $v = a_i$ *for* $i > c$ *every edge* $e$ *non-incident with edges from* $C$, *but incident with* $v$ *is a good edge. Assume by contradiction that this is not the case. Denote by* $\ddot{G}(A, B)$ *the graph obtained from* $G(A, B)$ *by deleting edges of* $C$. *For a subset* $S \subseteq A$ *we will denote by* $N(S)$ *the set of neighbors of the vertices of* $S$ *in* $\ddot{G}(A, B)$. *Graph* $\ddot{G}(A, B)$ *obviously has a perfect matching (a sub-matching of the perfect matching of* $G(A, B)$). *Our assumption on* $e$ *let's us deduce that, if we exclude from* $\ddot{G}(A, B)$ *an edge* $e$ *together with its*

*endpoints, then the graph obtained in such a way does not have a perfect matching. So, using Hall's theorem we can conclude that there exists $S_v^e \subseteq A$ such that $v \notin S_v^e$ and, furthermore, the following two statements hold: [e is incident with some vertex from $m(S_v^e)$] and $[N(S_v^e) \subseteq m(S_v^e)]$.*

*Without loss of generality write $S_v^e = \{a_{c+1}, ..., a_{c+l}\}$ for some $l > 0$. Write $\Delta = \sum_{i=1}^{l} \delta_{c+i}$. Consider a fixed vertex $a_{c+i}$ for $i = 1, ..., l$. Denote by $\mathcal{F}_i$ the set of the vertices of $m(S_v^e)$ adjacent to it and by $\mathcal{G}_i$ the set of the vertices from the set $\{b_1, ..., b_c\}$ adjacent to it. Denote by $\mathcal{D}_i$ the set of the neighbors of $a_{c+i}$ in $G(A, B)$. Note first that $\mathcal{D}_{c+i} = \mathcal{F}_i \bigcup \mathcal{G}_i$ for $i = 1, ..., l$. Otherwise there would exist a vertex in $S_v^e$ adjacent to some vertex $x \notin m(S_v^e)$ of $\ddot{G}(A, B)$. But that contradicts the fact that $N(S_v^e) \subseteq m(S_v^e)$. For a vertex $b_j \in \mathcal{G}_i$ we call a vertex $a_j$ a reverse of $b_j$ with respect to $i$. Note that by symmetry $(a_j, b_{c+i})$ is an edge of $G(A, B)$. For a given vertex $a_{c+i}$ for $i = 1, 2, ...l$ write $Rev_i = \{(a_j, b_{c+i}) : b_j \in \mathcal{G}_i\}$. Note that the sets $Rev_1, ..., Rev(l)$ are pairwise disjoint and besides $|Rev_1 \bigcup ... \bigcup Rev_l| = \sum_{i=1}^{l} |\mathcal{G}_i|$. Therefore we can conclude that there are at least $\sum_{i=1}^{l} |\mathcal{G}_i| + \sum_{i=1}^{l} |\mathcal{F}_i|$ edges nonadjacent to $v$ and with one endpoint in the set $m(S_v^e)$. Thus, from the fact that $\mathcal{D}_{c+i} = \mathcal{F}_i \bigcup \mathcal{G}_i$ for $i = 1, 2, ..., l$, we can conclude that there are at least $\sum_{i=1}^{l} |\mathcal{D}_{c+i}|$ edges nonadjacent to $v$ and with one endpoint in the set $m(S_v^e)$. Since an edge $e$ has also an endpoint in $m(S_v^e)$, we can conclude that altogether there are at least $\Delta + 1$ edges in $G(A, B)$ with one endpoint in the set $m(S_v^e)$. But this completes the proof since it contradicts the definition of $\Delta$.*

## 13 Appendix - worst case example of asymmetric matching

We here illustrate a worst-case type analysis for the asymmetric regular graph setting.

One can ask whether it is possible to prove some reasonable $k$-anonymity in the asymmetric case (the general $\delta$-regular asymmetric bipartite graphs) for every person, rather than the *all-but-at-most* guarantee of theorem 4.1? The answer is - no. In fact we can claim more. For every $\delta$ there exist $\delta$-regular bipartite graphs $G(A, B)$ with the following property: *there exists an edge $e$ of some perfect matching $M$ in $G(A, B)$ and a vertex $w \in A$ nonadjacent to $e$ such that in every perfect matching $M'$ in $G(A, B)$ that uses $e$, vertex $w$ is adjacent to an edge from $M$.*

So, in other words, it is possible that if the adversary is lucky and knows in advance a complete record of one person then he will reveal with probability 1 a complete record of some other person. Thus, those types of persons do not have much privacy. Fortunately, theorem 4.1 says that if the publisher chooses the parameters of a $\delta$-regular bipartite graph he creates carefully then there will only be a *tiny fraction* of persons like that. We next show constructions of asymmetric $\delta$-regular bipartite graphs for which the adversary, if given information about one specific edge of the matching in advance, can find another edge of the matching with probability 1.

For a fixed $\delta$ our constructed graph $G(A, B)$ will consist of color classes of sizes $\delta^2 + 1$ each. The graph $G(A, B)$ is the union of $\delta + 2$ bipartite subgraphs and some extra edges added between these graphs. The subgraphs are:

- subgraphs $F_i : i = 1, 2, ..., (\delta - 1)$, where each $F_i$ is a complete bipartite graph with one color class of size $\delta$ (the one from $A$) and one color class of size $(\delta - 1)$ (the one from $B$). We denote the set of vertices of $F_i$ from $A$ as $\{u_1^i, ..., u_\delta^i\}$ and from $B$ as $\{d_1^i, ..., d_{\delta-1}^i\}$
- bipartite subgraph $B_1$ of two adjacent vertices: $x \in A, y \in B$
- bipartite subgraph $B_2$ with vertex $z \in A$ and $k$ vertices from $B$ adjacent to it, namely $\{r_0, r_1, ..., r_{\delta-1}\}$
- complete $(\delta - 1)$-regular bipartite subgraph $B_3$ with color classes $\{w_1, ..., w_{\delta-1}\} \subseteq A$ and $\{v_1, ..., v_{\delta-1}\} \subseteq B$

with edges between the above $\delta + 2$ subgraphs as follows:

- $(y, u_1^i)$ for $i = 1, 2, ..., \delta - 1$
- $(r_{\delta-i}, u_j^i)$ for $i = 1, 2, ..., (\delta - 1); j = 2, 3, ..., \delta$
- $(x, v_i)$ for $i = 1, 2, ..., \delta - 1$
- $(r_0, w_i)$ for $i = 1, 2, ..., \delta - 1$.

Consider when an adversary attacks the above constructed graph $G(A, B)$ after knowing one edge in advance. It is enough to prove that any matching in $G(A, B)$ that uses $(x, y)$ must also use $(z, r_0)$. So assume by contradiction that there is a matching $M$ in $G(A, B)$ that uses both $(x, y)$ and $(z, r_{\delta-i})$ for some $i \in \{1, 2, ..., \delta - 1\}$. Denote by $\grave{G}(A, B)$ the graph obtained from $G(A, B)$ by deleting $x, y, z, r_{\delta-i}$. This graph must have a perfect matching. However it does not satisfy Hall's condition. The condition is not satisfied by the set $\{u_1^i, ..., u_\delta^i\}$ because one can easily check that in $\grave{G}(A, B)$ we have: $N(\{u_1^i, ..., u_\delta^i\}) = \{d_1^i, ..., d_{\delta-1}^i\}$. That completes the proof.