[Reviews · NeurIPS 2013]

Submitted by Assigned_Reviewer_3

This paper introduces a graph theoretic extension of the notion of k-anonymity, b-matching anonymity.
Correspondences of records before and after anonymization are represented as a bipartite compatibility graph, reffered to as a compatibility graph. If the graph is k-anonymized, the compatibility graph forms k-regular bipartite graph.
Existing k-anonimity requires that k-anonymous records should form a graph with fully-connected k-cliques.
The authors extend the notion of k-anonymity to b-matching anonymity so that the compatibility graph forms a k-regular bipartite graph.
This relaxation provides adaptive anonymity, in that each record is allowed to set the anonymity parameter independently with respect to the desired privacy revel .
The contribution of the authors are:
(1) polynomial algorithms to find b-maching anonymized records by alternate execution of the max-flow algorithm (if my understanding is correct) and minimization of a variational upper bound
(2)characterization of the b-maching anonymity against adversaries who know a limited number of edges
(3) empirical evaluation of the utility

The paper is well organized; notions and theorems are clearly stated.
Notions similar to b-matching anonymity are introduced by a couple of papers (e.g., [A] and [B]); relations between them should be mentioned in the paper.
Characterization of b-maching anonymity against adversaries with background knowledge seems to be quite new and opens up a new vista in this field.
I personally think this is a paper for STOC or FOCS and I am not sure if NIPS readers are interested in this paper.

[A]Mingqiang Xue, Panagiotis Karras, Chedy Raïssi, Jaideep Vaidya, Kian-Lee Tan: Anonymizing set-valued data by nonreciprocal recoding. KDD 2012: 1050-1058
[B]Sachin Lodha, Dilys Thomas: Probabilistic Anonymity. PinKDD 2007: 56-79
Summary: Characterization of b-maching anonymity against adversaries with background knowledge seems to be quite new and opens up a new vista in this field.
I personally think this is a paper for STOC or FOCS and I am not sure if NIPS readers are interested in this paper.

Submitted by Assigned_Reviewer_6

The k-anonymity is a well known concept. The authors suggest a novel extension of it - b-matching. They analyze the difference between the two and devise a novel algorithms to generate efficient solutions for it.

Comments:
Line 020: remove the comma after "and"
Line 057: ...entry y_i of Y...
Line 061: at least k-1

Suggestion: you can use different weights for different features (one can be more willing to disclose his age than the fact that he is HIV positive).
Summary: The paper defines a new measure for privacy, and shows how to efficiently anonymize data set accordingly.

Submitted by Assigned_Reviewer_7

'Adaptive Anonymity via b-Matching' addresses the problem of releasing a private dataset such that by censoring measurements individuals in the dataset become indistinguishable. The authors explain the concepts of k-anonymity and the b-matching problem as forming a bipartite graph that describes the compatibility of the original dataset with the released dataset. They provide two algorithms, for the asymmetric and symmetric cases, which are justified as variational approximations of the star (suppression) minimization problem. There is an approximation result for the algorithms. A variety of privacy guarantees are provided for single attacks,and sustained attacks on the k-clique construction, symmetric and asymmetric b-matchings. A brief experimental section is provided with some convincing results.

This paper is of high quality, as it comprehensively addresses the b-matching problem and provides strong computationally feasible algorithms that are well justified. It is very well written with a very understandable introduction and exposition. The algorithmic results are strong, as are the privacy guarantees. It is likely to be a significant work, as b-matching is a significant extension of the k-anonymity, while being a reasonable alternative to other notions of privacy. Furthermore, this work is very general, allowing for user specified levels of privacy and addressing both the symmetric and asymmetric cases. The only criticism is that theorem 1 is quite weak as an approximation factor of the degree makes it nearly useless. I think that an explanation (at least in the appendix) of how the algorithms are a variational upper bound of the ideal program would be more convincing.

Minor comments:
191 - 'a more general' -> 'a general'
263 - use `` not ''
def 4.1 - this is confusing, how does the matching come in to the symmetry notion here
Summary: This paper comprehensively addresses the b-matching problem for k-anonymous privacy, providing strong privacy guarantees and sophisticated novel algorithms. I strongly recommend its acceptance as it promises to be an important study of k-anonymity through b-matching.
Author Feedback

Author rebuttal: We thank the reviewers for their time and their helpful comments. We address each reviewer by the ID that appears on our review page (3, 6, 7).

== Reviewer 3 ==
We thank Reviewer 3 for bringing references [A] and [B] to our attention. We will mention these in the revised paper. The nonreciprocal recoding of [A] is indeed our asymmetric b-anonymity. However, our adaptivity and the symmetric extension are novel. Particularly relevant is our result that while symmetric b-anonymity is as secure as k-anonymity, asymmetric b-anonymity is weaker against sustained attacks. This is a point that [A] ignores; in fact, they claim “reciprocity assumption is not required by a privacy condition; it is redundant.” Reference [B] relaxes k-anonymity probabilistically; in general, each record is required to be identical with k others. This is not directly comparable to our b-anonymity relaxation. Moreover, their method requires a generalization hierarchy of attribute values.

Next, the reviewer asks whether NIPS readers would be interested in our paper. We believe they will: NIPS has accepted 8 privacy related papers in the last 5 years [1-8], with 3 last year alone. Some have been highly cited: the landmark [1] by 61, [4] by 38, and [5] and [8] each by 15, according to a Google Scholar search. This evidence suggests that (1) the NIPS audience is interested in privacy, particularly since last year; and (2) NIPS is a high-impact venue for privacy works.

We believe that our work can have a high impact in this part of the NIPS community because our method complements the majority of existing work that focuses on differential privacy, for which restrictions on the learning task must be specified in advance. Our method is agnostic to the learning task.

While the paper presents several theorems regarding adaptive anonymity, NIPS is still a better fit than either STOC or FOCS. First, our experimental results are important to demonstrate that relaxing k-anonymity has practical value: the UCI results show that even without adaptivity, we improve utility; the social network results show that adaptivity adds even more utility. Moreover, there are gaps in our theory which we rely on experiments to address. First, our theory characterizes the privacy lost by going to the bipartite case but not the utility gained. Second, we wanted to empirically validate our iterative variational algorithm, beyond the result of Theorem 1

Moreover, though our theorems are new and important from the point of view of privacy, we are not trying to generate new theoretical tools for the broad theory literature. Our proofs build upon techniques for chains and posets (Dilworth's Theorem, Hall’s Theorem) in complicated and novel ways. FOCS and STOC, however, require altogether new theoretical proof techniques or solutions to long-standing theoretical problems, which is not our goal. Also, theory is also no reason to reject from NIPS; last year’s conference accepted a purely theoretical paper (no algorithms or experiments) on privacy aware learning [3], and related work [5, 7] also feature technical proofs relying on cryptography.

== Reviewer 6 ==
We thank Reviewer 6 for suggesting the HIV example to motivated weighted features. We mention weights in a footnote in lines 268–9 and can move it to the main text. To weight features, simply multiply W element-wise by another matrix, C. C can parameterize a weight for each user and feature. The higher the weight, the less likely an entry will be suppressed. A user can thus set the HIV weight low and the age weight high. Then the algorithm will prefer to suppress age rather than HIV.

== Reviewer 7 ==
We thank Reviewer 7 for their encouragement and suggestions. The degree bound in Theorem 1 is admittedly weak, but all known PTAS bounds for k-anonymity also depend on the degree [1 in paper]. We are working on a a stronger convergence bound for the variational algorithm. However, our experimental results show that in practice, the variational algorithm converges to a much better value than the theorem implies.

Our algorithms are variational because they maintain a concave upper bound through Jensen’s inequality. This is clarified in the proof in the supplemental material. The variational parameter is W, and for each value of W, we have a function of G, which upper bounds the star count s(G). Thus, alternately minimizing G (a weighted matching problem) and W (analytic) maintains the upper bound.

References
[1] K. Chaudhuri and C. Monteleoni. Privacy-preserving logistic regression. In NIPS, pages 289–296, 2008.
[2] K. Chaudhuri, A. Sarwate, and K. Sinha. Near-optimal differentially private principal components. In NIPS, pages 998–1006. 2012.
[3] J. Duchi, M. Jordan, and M. Wainwright. Privacy aware learning. In NIPS, pages 1439–1447. 2012.
[4] M. Hardt, K. Ligett, and F. McSherry. A simple and practical algorithm for differentially private data release. In NIPS, pages 2348–2356, 2012.
[5] M. Kearns, J. Tan, and J. Wortman. Privacy-preserving belief propagation and sampling. In NIPS, 2007.
[6] J. Lei. Differentially private m-estimators. In NIPS, pages 361–369, 2011.
[7] M.A. Pathak, S. Rane, and B. Raj. Multiparty differential privacy via aggregation of locally trained classifiers. In NIPS, pages 1876–1884, 2010.
[8] O. Williams and F. McSherry. Probabilistic inference and differential privacy. In NIPS, pages 2451–2459, 2010.